# Worse than Zero-shot?
# A Fact-Checking Dataset for Evaluating the Robustness of RAG Against Misleading Retrievals

**Linda Zeng**[*]
The Harker School
San Jose, California, USA
lindazeng979@gmail.com

**Rithwik Gupta**[*]
Irvington High School
Fremont, California, USA
rithwikca2020@gmail.com

**Divij Motwani**
Palo Alto High School
Palo Alto, California, USA
divijmotwani@gmail.com

**Yi Zhang**[†]
University of California Santa Cruz
Santa Cruz, California, USA
yiz@ucsc.edu

**Diji Yang**[†]
University of California Santa Cruz
Santa Cruz, California, USA
dyang39@ucsc.edu

## Abstract

Retrieval-augmented generation (RAG) has shown impressive capabilities in mitigating hallucinations in large language models (LLMs). However, LLMs struggle to maintain consistent reasoning when exposed to misleading or conflicting evidence, especially in real-world domains such as politics, where information is polarized or selectively framed. Mainstream RAG benchmarks evaluate models under clean retrieval settings, where systems generate answers from gold-standard documents, or under synthetically perturbed settings, where documents are artificially injected with noise. These assumptions fail to reflect real-world conditions, often leading to an overestimation of RAG system performance. To address this gap, we introduce RAGUARD, the first benchmark to evaluate the robustness of RAG systems against *misleading* retrievals. Unlike prior benchmarks that rely on synthetic noise, our fact-checking dataset captures naturally occurring misinformation by constructing its retrieval corpus from Reddit discussions. It categorizes retrieved evidence into three types: *supporting*, *misleading*, and *unrelated*, providing a realistic and challenging testbed for assessing how well RAG systems navigate different types of evidence. Our experiments reveal that, when exposed to potentially misleading retrievals, all tested LLM-powered RAG systems perform worse than their zero-shot baselines (i.e., no retrieval at all), while human annotators consistently perform better, highlighting LLMs' susceptibility to noisy environments. To our knowledge, RAGUARD is the first benchmark to systematically assess the robustness of the RAG against misleading evidence. We expect this benchmark to drive future research toward improving RAG systems beyond idealized datasets, making them more reliable for real-world applications.[1]

## 1 Introduction

Retrieval-augmented generation (RAG) systems show strong potential for mitigating large language model (LLM) hallucinations and enhancing trustworthiness. By combining the generative capabilities

---

[*]Equal contribution.

[†]Co-advising.

[1]The dataset is available at https://huggingface.co/datasets/UCSC-IRKM/RAGuard.

39th Conference on Neural Information Processing Systems (NeurIPS 2025) Track on Datasets and Benchmarks.

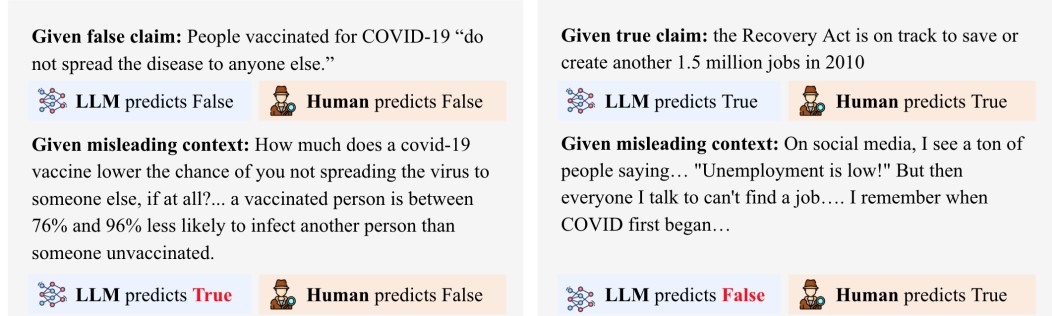

Figure 1: Examples of LLM and human performance on a false claim (left) and a true claim (right) from RAGUARD. While the LLM initially classified both claims correctly, it later reversed its decisions due to misleading retrieved context. In contrast, human judgments remained consistent.

of LLMs with the retrieval power of external corpora, RAG aims to ground responses in relevant information, thus improving factual consistency and output credibility [21, 13, 9]. While existing work has made significant progress in improving retrieval relevance and maximizing the amount of information in the retrieved context [17, 43, 5], comparatively less attention has been paid to scenarios where LLMs must reason over inevitably misleading, conflicting, or only partially relevant retrieved content, as illustrated in Figure 1. Addressing this robustness gap is increasingly important as RAG systems are deployed in high-stakes applications such as fact-checking [32] and legal or medical domains [11, 41].

Prior work has mitigated LLMs' susceptibility to noisy retrievals by prompting models to evaluate each retrieved document's relevance [37], isolating LLMs' responses to individual passages before aggregating them [39], or prompting agents to select external knowledge based on a debate process [35]. However, these approaches largely focus on filtering or restructuring retrieval rather than tackling the core challenge of LLM reasoning over misleading information. Many approaches aim to reconcile temporal or factual inconsistencies between retrieved content and an LLM's prior knowledge rather than addressing cases where the retrieved information itself is misleading or contradictory [34, 18]. Furthermore, current datasets overly rely on curating reliable documents, limiting robustness testing against misinformation [16, 44, 26, 20]. While some introduce counterfactuals or retrieval noise [22, 6], they rely on artificial perturbations or costly human annotation. This highlights the need for an evaluation framework that challenges RAG systems with naturally occurring contradictions, as well as a taxonomy of evidence types that clarifies their distinct impacts on model behavior.

Fact-checking plays a crucial role in combating misinformation, yet most existing datasets in this domain assume the availability of gold-standard evidence aligning with the verdict [33, 2, 45, 14, 19, 4]. This assumption breaks down in political domains, where controversial claims lead to both supporting and opposing narratives from diverse sources [23, 33, 25, 29]. To build fact-checking systems capable of handling real-world misinformation, it is essential to expose models to the conflicting and misleading evidence with which humans work in the real world.

To bridge this gap, we introduce RAGUARD, a benchmark dataset based on political claims and their verifications from PolitiFact that incorporates real-world misinformation. Given the prevalence of polarizing and deceptive information in political discourse, we develop an automated pipeline that retrieves relevant yet potentially misleading user-generated content and labels them through a novel LLM-guided approach simulating a fact-checking exam. We then evaluate widely used LLM-based RAG systems, confirming that current methods lack robustness in real-world scenarios. Performance drops significantly when RAG systems are exposed to documents from the RAGUARD knowledge base, revealing a substantial gap from human reasoning in identifying and handling misleading evidence.

In summary, our work advocates for a shift from idealized RAG settings to ones that reflect the misleading nature of real-world retrieval. We introduce a taxonomy, benchmark, and evaluation framework for assessing LLM robustness under such conditions. Our contributions are as follows:

- **Task:** We define a new robustness-focused fact verification task that challenges models to reason through misleading retrieved content. We also unify inconsistent terminology in prior work by establishing a structured framework for labeling document types (e.g., supporting, misleading, unrelated).

- **Benchmark:** We release RAGUARD, a real-world political-domain RAG benchmark built from PolitiFact claims and Reddit retrievals. Documents are labeled based on their effect on LLM predictions using a scalable, LLM-guided annotation method targeting misleading effect on models rather than humans.

- **Evaluation:** We evaluate strong closed- and open-source models across multiple retrieval settings, revealing that even top-performing LLMs struggle under misleading context with all models performing worse than their zero-context baselines.

## 2   Dataset

We introduce RAGUARD, a benchmark for evaluating the robustness of RAG systems in political fact-checking. In the following sections, we standardize terminology used in prior work (Section 2.1), compare RAGUARD to existing datasets (Section 2.2), describe its structure and key statistics (Section 2.3), and outline the fact-checking tasks it supports (Section 2.4).

### 2.1   Task and Terminology

The core task in RAGUARD is retrieval-augmented fact-checking: determining whether a claim is true or false based on retrieved evidence that may be *supporting*, *misleading*, or *unrelated*. Unlike prior datasets that include only documents explicitly supporting the correct answer, RAGUARD adopts a broader definition of *supporting*, allowing documents that provide contextual cues even if they do not state the answer outright.

To better reflect real-world retrieval conditions, we introduce noise through *unrelated* documents, which are topically related but uninformative, and crucially, *misleading* documents, which subtly distort facts through framing, omission, or biased presentation. Unlike adversarially fabricated content or clearly one-sided, unambiguous evidence, misleading noise arises naturally and rarely contains explicit falsehoods; instead, it subtly presents facts or context in ways that are misleading to models but often recognizable to humans (see Figure 1). Importantly, the quality of being misleading in RAGUARD is defined relative to language models rather than as an objective property discernible by all humans. The dataset is designed to expose specific vulnerabilities in model reasoning, cases where LLMs fail to separate factual content from bias or rhetorical tone. Thus, the task reflects the more realistic challenge of distilling truth from partial or polarized viewpoints, as human fact-checkers must do, rather than merely answer matching with given evidence.

Finally, we unify terminology from prior work to clarify distinctions between evidence types. While some documents may be non-conflicting but distracting (e.g., unrelated or randomly-selected), others are conflicting and more challenging, such as misleading, fabricated, or unambiguous evidence. We illustrate this hierarchical structure of definitions in Figure 2 and explicitly define all document types in our taxonomy (see Appendix A), which helps situate RAGUARD within the broader space of retrieval-augmented fact-checking datasets.

### 2.2   Comparison with Existing Datasets

**Fact-Checking Datasets.**   Table 1 summarizes key properties of existing fact-checking and RAG benchmarks, including whether retrieval is used, conflicting evidence is included, and documents are drawn from real-world sources. Most fact-checking datasets that incorporate evidence retrieval are limited to supporting documents and do not account for conflicting or misleading information. For example, while FEVEROUS [2] categorizes some evidence as *refuted*, this label only applies to documents that help the model correctly classify a claim as false, not those that contradict the fact-checking verdict. Additionally, both FEVER [33] and FEVEROUS rely on curated, rewritten Wikipedia passages, rather than naturally occurring claims or user-generated content [4].

While Liar [36] and Mocheg [45] also source claims from PolitiFact, Liar does not support evidence retrieval, and Mocheg includes only gold-standard documents cited by PolitiFact fact-checkers.

| Dataset | Focus | | Evidence | | | Claims | |
|---|---|---|---|---|---|---|---|
| | FC | ROB | Retrieval | Conflicting | Real-world | Domain | # Claims |
| FEVER [33] | ✓ | ✗ | ✓ | ✗ | ✗ | General | 185K |
| FEVEROUS [2] | ✓ | ✗ | ✓ | ✗ | ✗ | General | 87K |
| Liar [36] | ✓ | ✗ | ✗ | ✗ | ✓ | Political | 12.8K |
| Mocheg [45] | ✓ | ✗ | ✓ | ✗ | ✓ | Political | 15.6K |
| Snopes [14] | ✓ | ✗ | ✓ | ✗ | ✓ | Political | 6.4K |
| PubHealth [19] | ✓ | ✗ | ✓ | ✗ | ✓ | Health | 11.8K |
| MultiFC [4] | ✓ | ✗ | ✓ | ✗ | ✓ | Political | 43.8K |
| AVeriTeC [28] | ✓ | ✗ | ✓ | ✓ | ✓ | Political | 4.6K |
| Power of Noise [6] | ✗ | ✓ | ✓ | ✗ | ✓ | General | 10K |
| RAAT [8] | ✗ | ✓ | ✓ | ✓ | ✗ | General | 7.8K |
| NoiserBench [38] | ✗ | ✓ | ✓ | ✓ | ✗ | General | 4K |
| QACC [22] | ✗ | ✓ | ✓ | ✓ | ✓ | General | 1.5K |
| RAGUARD (ours) | ✓ | ✓ | ✓ | ✓ | ✓ | Political | 2.6K |

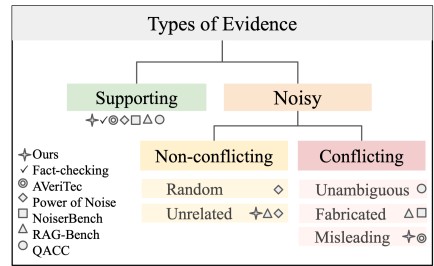

Table 1: Comparison of RAGUARD with fact-checking and noisy RAG datasets. "FC" indicates suitability for fact-checking, and "ROB" for LLM robustness evaluation. Columns reflect evaluation focus, evidence types, and dataset characteristics.

Figure 2: Taxonomy of document types used in our benchmark (supporting, misleading, unrelated), along with types of evidence included in related datasets.

Similarly, other datasets [14, 19, 4] primarily use journalist-written explanations from fact-checking websites, which are explicitly curated to justify the verdict. In contrast, RAGUARD incorporates conflicting evidence from naturally occurring Reddit discussions, reflecting more realistic challenges.

AVeriTeC [28] also explores real-world misleading information, focusing on retrieval-augmented fact verification with naturally occurring conflicting evidence. Like RAGUARD, their dataset includes claims and evidence drawn from real-world sources, some of which subtly distort the truth through biased framing or misinformation. However, a key difference lies in the task framing: their benchmark treats evidence as "conflicting" when it contains both supporting and refuting signals and asks the model to abstain from a definitive verdict in such cases. In contrast, RAGUARD explicitly expects the model to reason *through* misleading evidence and arrive at a correct verdict. While AVeriTeC focuses on detecting ambiguity in the evidence, RAGUARD challenges models to exhibit robustness under noisy, real-world conditions. As such, our task is not only stricter but also aligned with practical fact-checking requirements.

**Datasets with Noisy Contexts.** Several prior datasets introduce noisy contexts for evaluating retrieval-augmented generation, primarily in open-domain QA settings [22, 6, 38, 8]. However, these datasets vary significantly in how they define noise and the types of disruptions they model. Power of Noise [6] introduces noise through unrelated documents, in the form of off-target retrievals or random documents entirely irrelevant to the query. Importantly, the dataset does not introduce any content that actively contradicts or distorts the correct answer (i.e., its noise is exclusively non-conflicting.) RAAT [8] introduces counterfactual noise by editing documents to contain incorrect answers. While this creates explicit conflicting evidence, it does so synthetically, often resulting in unrealistic or adversarial examples that lack subtlety (e.g., "Titanic earned a worldwide total of 2.187 billion" is directly replaced with a different number, which would likely mislead not only an LLM but also a human). Similarly, the noise in NoiserBench [38] is fabricated, making it poorly representative of how misleading evidence actually appears in public discourse. QACC [22] uses human annotators to label naturally retrieved documents as conflicting or not, avoiding artificial edits. However, the definition of conflict in QACC is binary (i.e., whether the document directly supports or contradicts the gold answer) leaving little room for subtler forms of misleading reasoning (see Figure 1).

RAGUARD complements these datasets by focusing specifically on real-world political claims, including user-generated texts containing naturalistic misinformation that challenge LLMs with plausible distortions rather than clear factual opposition. Unlike prior work, which avoids conflicting content or reduces it to fabricated or unambiguous contradictions, RAGUARD expects models to reason through *misleading* evidence and infer the correct label, reflecting real-world fact-checking scenarios, where misinformation is embedded in discourse rather than stated outright.

## 2.3 Dataset Structure

RAGUARD consists of 2,648 political claims made by U.S. presidential candidates (2000–2024), each labeled as either *true* or *false*, and a knowledge base comprising 16,331 documents. Figure 3a presents

| Statistic | Value |
|---|---|
| Total Claims | 2,648 |
|   True | 1,333 (50.3%) |
|   False | 1,315 (49.7%) |
| Avg. Claim Length | 17.6 words |
| Total Documents | 16,331 |
|   Supporting | 2,685 (16.4%) |
|   Misleading | 1,812 (11.1%) |
|   Unrelated | 11,834 (72.5%) |
| Avg. Doc Length | 161 words |
| Avg. Docs/Claim | 6.2 |
| Claims w/ Supporting Docs | 955 (36.1%) |
| Claims w/ Misleading Docs | 788 (29.8%) |

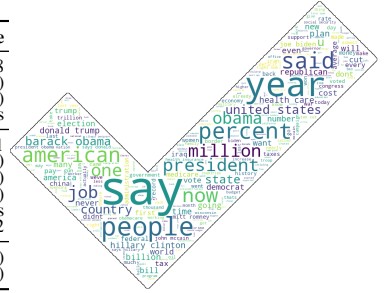
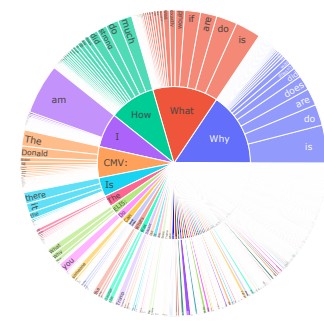

(a) Key statistics on claims and documents, including class balance and average lengths.

(b) Frequent words in claims, shaped as a checkmark to reflect verification focus.

(c) Common opening words in documents, with many thin segments indicating high diversity.

Figure 3: Overview of RAGUARD, including dataset statistics and word frequencies.

the key statistics of the dataset. Each claim is linked to a set of associated documents, categorized as *supporting*, *misleading*, or *unrelated*, with an average of 6.2 documents per claim. Notably, the dataset contains more supporting documents than misleading ones, reflecting that political discussions online are more often aligned with factual information, while the large number of unrelated documents suggests that many discussions online are neutral, neither misleading nor supporting the validity of a claim. Appendix B provides additional statistics related to the year and speaker of each claim.

RAGUARD includes a diverse and realistic collection of political claims and documents. Figure 3b visualizes the most frequent words in claims, revealing a focus on reported assertions (e.g., "say") and quantitative language (e.g., "percent," "million") that require quantitative reasoning. The frequent occurrence of temporal terms like "year" further indicates that many claims are time-sensitive, requiring temporal awareness that may challenge both human fact-checkers and LLMs.

Figure 3c illustrates the lexical diversity of the retrieval corpus. The inner ring denotes the first word in each document's opening sequence, while the outer ring shows the subsequent word. The abundance of narrow, evenly distributed segments demonstrates that no single phrase or construction dominates the corpus. This diversity helps prevent models from exploiting superficial lexical cues, ensuring that success depends on genuine reasoning rather than memorized linguistic patterns. Notably, many retrieved documents begin with questions, mirroring the exploratory and uncertain tone of real-world online discussions where factuality is often debated rather than asserted.

## 2.4 Supported Tasks

To benchmark the performance of current RAG systems in real-world fact-checking scenarios, we define a series of tasks using RAGUARD.

**Zero-Context Prediction.** This task assesses the model's ability to fact-check claims using only its internal knowledge, with no retrieved documents. It serves as a zero-shot baseline for evaluating the impact of retrieval.

**Standard RAG.** This task requires RAG systems to retrieved documents from the entire dataset corpus in real time. Retrieved context may include ground-truth documents related to the claim (unrelated, supporting, or misleading information) or unrelated documents, simulating noisy retrieval in real-world settings.

**Oracle Retrieval.** This task isolates the effect of retrieval content quality by bypassing real-time search and directly supplying documents known to be associated with a given claim. We evaluate two conditions: In the first, which we denote as Oracle Retrieval (All), the model receives a document labeled as supporting, misleading, or unrelated, testing its ability to reason over mixed or ambiguous evidence. In the second, which we refer to as Oracle Retrieval (Misleading), the model is exposed only to documents that conflict with the claim's ground-truth label, providing a targeted evaluation of susceptibility to deceptive or adversarial content.

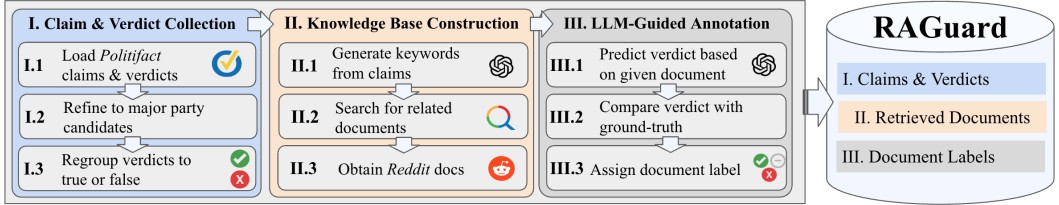

Figure 4: RAGUARD dataset construction, consisting of three stages to obtain claims and verdicts, associated documents, and labels for the each document's relationship to the claim and verdict.

# 3 Dataset Construction

RAGUARD is constructed in three stages, as shown in Figure 4: (1) collecting political claims and verdicts, (2) retrieving documents from Reddit, and (3) labeling documents as *supporting*, *misleading*, or *unrelated* using an LLM-guided protocol. Our key novelty lies in stage (3), where we define being misleading relative to model behavior rather than human judgment, allowing RAGUARD to scale beyond human annotation and target LLM-specific weaknesses.

## 3.1 Claim and Verdict Collection

We scrape political claims and verdicts from PolitiFact,[2] a professional fact-checking organization. To ensure claims are both prominent and controversial enough to spur misinformation, we restrict our dataset to US presidential candidates from 2000 to 2024. For clarity in downstream document retrieval and annotation, we binarize PolitiFact's six-point truth scale (*true, mostly true, half true, mostly false, false, pants on fire*) into *true* and *false* verdicts, omitting intermediate categories as it is challenging for a document to specifically mislead a *half true* verdict.

## 3.2 Knowledge Base Construction

To simulate real-world retrieval noise, we construct a retrieval corpus from Reddit,[3] a platform rich with user-generated political discussion and misinformation. Because Reddit users inherently publish their opinions to the public, posts are often written to be convincing, regardless of how true the information is. For each claim, we use GPT-4 to generate keyword variants, increasing the likelihood of retrieving diverse types of information related to the topic. We then issue a Google Search restricted to Reddit and collect the top ten retrieved posts per claim. This keyword-based search ensures contextual relevance, while Reddit's user-generated content introduces diverse perspectives, including both speculative theories and well-supported arguments, mirroring the complexities of real-world fact-checking challenges.

## 3.3 LLM-Guided Annotation

We annotate each retrieved document based on its functional effect on the model's prediction. Rather than relying on manual judgments or synthetic counterfactuals, we simulate the document's use in a RAG pipeline by prompting GPT-4 to classify the claim using only the document as context. We then compare its prediction to the ground-truth verdict. If the document helps the model arrive at the correct label, it is labeled *supporting*. If the document causes an incorrect prediction, it is labeled *misleading*. If the model considers the document unrelated to its prediction, it is labeled *unrelated*. Appendix C.1 includes prompts for keyword extraction and LLM annotation.

This LLM-as-annotator strategy enables scalable, behavior-based labeling. Because our dataset aims to reveal weaknesses in how LLMs interpret biased or conflicting information, we define misleadingness relative to model behavior (i.e., by a document's ability to confuse the model rather than by human interpretation). Human annotators can typically resolve these cases with ease (Section 5). Moreover, these labels are used solely as an intermediate analytical tool to characterize how retrieved evidence influences model behavior. While models may encounter incorrect or

---

[2]https://www.politifact.com
[3]https://www.reddit.com

| | Open Source | | | Closed Source | | | Reasoning | |
|---|---|---|---|---|---|---|---|---|
| | OLMo-1B | Llama 3 | Mistral | Gemini 1.5 | GPT-4o | Claude 3.5 | DeepSeek | o4-mini |
| Zero-Context Prediction | 56.87 | 62.50 | 63.97 | 61.06 | 67.33 | 74.51 | 69.98 | 63.67 |
| RAG-1 | 52.68 | 59.40 | 59.14 | 56.68 | 64.80 | 70.09 | 66.88 | 62.76 |
| RAG-5 | 49.74 | 61.37 | 58.91 | 57.59 | 65.90 | 68.58 | 57.81 | 63.14 |
| Oracle Retrieval (All) | 53.89 | 61.09 | 51.55 | 52.38 | 53.22 | 52.56 | 50.06 | 51.88 |
| Oracle Retrieval (Misleading) | 44.04 | 36.81 | 26.88 | 30.57 | 45.97 | 35.98 | 38.25 | 33.39 |

Table 2: Accuracy (%) of various LLM backbones in RAG setup across three tasks and five evaluation settings. Cell color intensity corresponds to the model's percent accuracy drop relative to its zero-context baseline, with darker red indicating larger relative performance drop. Appendix D contains a version of Table 2 with exact relative percent decreases shown.

misleading information in the retrieved context, the task evaluates their ability to verify claims against gold verdicts from PolitiFact, rather than to detect misinformation within the documents themselves.

To confirm this process does not overfit to GPT-4 idiosyncrasies, we re-annotated a subset of the dataset with Claude 3.5 Sonnet and Gemini 1.5 Flash and measured inter-annotator agreement using Cohen's $\kappa$. We observe substantial agreement with Claude 3.5 Sonnet ($\kappa = 0.789$) and moderate agreement with Gemini 1.5 Flash ($\kappa = 0.650$) on the supporting and misleading labels. These results indicate that the labeling decisions are reasonably stable across models rather than solely an artifact of GPT-4's annotation behavior. As shown in Section 4, multiple LLMs exhibit similar vulnerabilities on the same misleading documents, reinforcing the generality of our findings.

# 4 Baselines

## 4.1 Experimental Setup

**Evaluation.** We frame fact-checking as a binary classification task where the model must generate a response that aligns with one of the predefined options. Accuracy, calculated using the ground-truth verdict, is used to evaluate performance. If a model generates an out-of-scope response that does not match any of the given options, it is treated as an incorrect prediction.

**Implementation Details.** We evaluate eight LLMs: three open-source models at different scales (OLMo-1B [10], Llama 3 8B Instruct [7], and Mistral 7B Instruct [15]), three commercial APIs (Gemini 1.5 Flash [31], GPT-4o [1], and Claude 3.5 Sonnet [3]), and two closed-source reasoning-oriented models (DeepSeek R1 [12] and o4-mini [24]). For the Standard RAG setting, we perform retrieval using OpenAI's `text-embedding-ada-002` model, retrieving the top one (RAG-1) and five (RAG-5) documents based on semantic similarity to the claim. In the Oracle Retrieval setting, we directly supply the pre-labeled associated documents without performing retrieval. All prompts indicate that provided context may be unrelated or factually incorrect (see Appendix C.2).

In addition to standard baselines, we evaluate a robustness-oriented RAG method, Corrective RAG (CRAG) [42]. We reproduce CRAG using the authors' released code, prompts, and pretrained Critic. All experiments are conducted under our Oracle Retrieval setting, where CRAG receives pre-labeled documents from RAGUARD. We compare their Llama 2-based system to zero-shot Llama 2.

## 4.2 Results

Table 2 displays baseline results on RAGUARD for three tasks using three open-source, three closed-source, and two closed-source reasoning LLMs.

**Zero-Context Prediction.** All systems achieve the highest accuracy on the zero-context prediction (i.e., zero-shot baseline), which is counterintuitive, considering this setting does not benefit from retrieval. Furthermore, we find that reasoning models do not achieve higher zero-context prediction scores since the zero-context task relies primarily on prior knowledge rather than reasoning capability.

**Standard RAG.** Adding retrieved context consistently reduces performance across all models. While o4-mini is the most robust, the performance of models like Mistral and Gemini 1.5 drops sharply.

More retrieved documents (RAG-5) often worsen performance compared to RAG-1, especially for stronger models like Claude and DeepSeek, suggesting that retrieval introduces primarily distracting and misleading information, and when the quality of retrieved context is not optimal, including more documents can confuse rather than help, particularly if the model already performs well on the task. These findings challenge the assumption that retrieval improves accuracy and align with concerns about retrieval quality in real-world tasks [27, 46, 6, 40].

**Oracle Retrieval.** In the Oracle Retrieval (All) condition, where models receive all documents explicitly associated with each claim, performance drops even further compared with both Standard RAG and the zero-shot baseline. The effect is most severe in the Oracle Retrieval (Misleading) condition, which yields an average accuracy decrease of 46.5%. Every model falls below 50% accuracy despite the binary nature of the task, confirming that the misleading evidence in RAGUARD explicitly disrupts model reasoning. These results demonstrate that current RAG systems are unable to distinguish factual content from subtle misinformation.

**Existing Robustness Method.** Despite being designed to mitigate retrieval errors, the CRAG method performs substantially worse than the zero-shot baseline: Llama 2 achieves 50.57% without retrieval but drops to 37.24% with CRAG in the Oracle Retrieval Setting. This method uses a lightweight evaluator to score document relevance and, when confidence is low, triggers web searches outside the retrieval corpus, primarily drawing from sources such as Wikipedia. In practice, this mechanism activates in 70.1% of RAGUARD cases, but the added web content often introduces additional noise, as general-purpose sources such as Wikipedia provide limited support for verifying complex political claims. When the evaluator's confidence is low, CRAG combines web search results with documents from the corpus, effectively doubling exposure to noisy content. Therefore, while CRAG can detect the challenge of the problem, it remains unable to resist the misleading information. More broadly, this underscores that existing robustness methods, which have been shown to handle certain types of noise, remain vulnerable to the qualitatively different challenge of real-world misleading retrieval.

## 4.3 Analysis

**Model Comparison.** Across models, GPT-4o is the most robust overall: its accuracy falls by only 31.7% in the Oracle Retrieval (Misleading) condition, compared to Claude 3.5 (51.8%), Gemini 1.5 (49.9%), and Mistral (58.0%). This pattern seems counterintuitive given that parts of the dataset were shaped by GPT-4's own failure modes. Nevertheless, its robustness in this setting suggests that the annotation process does not overly capture GPT-4-specific failure patterns and that its high performance may instead be bolstered by its inherent reasoning strength, consistent with recent studies showing GPT-4's superior fact-checking capabilities [30], as further discussed in Appendix D.1. In contrast, Claude 3.5 achieves the highest zero-context accuracy but suffers the steepest decline, suggesting that strong internal knowledge does not guarantee resistance to misleading information.

Reasoning-focused systems such as o4-mini and DeepSeek show large performance drops of 45.3% and 47.6%, respectively, and smaller models like OLMo-1B follow the same pattern, with all RAG variants underperforming their zero-shot baselines. These consistent trends across model size and training type underscore that RAGUARD probes weaknesses beyond those captured by conventional reasoning or robustness benchmarks.

**Retrieval and Misleading Evidence.** The fact that Oracle Retrieval leads to worse performance than Standard RAG suggests that retrieval errors, while suboptimal, may result in less damaging content than intentionally misleading documents, validating the construction and annotation of such documents in RAGUARD. To quantify this phenomenon, we introduce Misleading Retrieval Recall, which measures the proportion of claims for which at least one misleading document is retrieved. In RAG-1, this rate is 21.3%, increasing to 44.8% in RAG-5, indicating that retrieving more documents raises the likelihood of including harmful content. We find that when Misleading Retrieval Recall is higher, as in Oracle Retrieval (Misleading), where Misleading Retrieval Recall is 100%, the LLM performance decreases further, demonstrating the more damaging effect of RAGUARD's misleading documents. Additional retrieval metrics are reported in Appendix D.2.

**Qualitative Examples.**    Figure 1 illustrates how misleading evidence interferes with LLM reasoning on specific claims. In the left example, the document provides an opinion questioning the claim without directly contradicting it (e.g., "less likely" does not invalidate the absoluteness of "anybody else"), yet the model misinterprets this subjective tone as factual evidence, a pattern we term confusing opinion with fact.

In the right example, the misleading document introduces contradictory information, but the discrepancy can be resolved by temporal reasoning: The claim and the evidence refer to different time periods ("2010" versus "when COVID first began"). We observe that models frequently fail to make such contextual distinctions. By misapplying contextual cues, they overemphasize superficial signals like numbers or names while ignoring the broader meaning. These examples reveal how LLMs overly focus on surface-level indicators, explaining why they are highly susceptible to misleading retrievals, even when humans can resolve the ambiguity with relative ease. We provide additional examples of these failures in Appendix E.

## 5    Human Study

To better understand how noisy context contained in RAGUARD affect reasoning, we study human robustness to noisy contexts to compare to LLM performance. We construct a 64-instance subset balanced across true and false claims, reflecting the distribution of misleading and supporting documents in our full dataset, while downsampling unrelated documents. On this subset, all models perform within 5% of the performance reported in Table 2. Of four recruited human annotators, two are PhD-level *experts* in adjacent fields and two self-identify to know minimal information about politics beyond the basics (*normal*). Annotator instructions are provided in Appendix F.

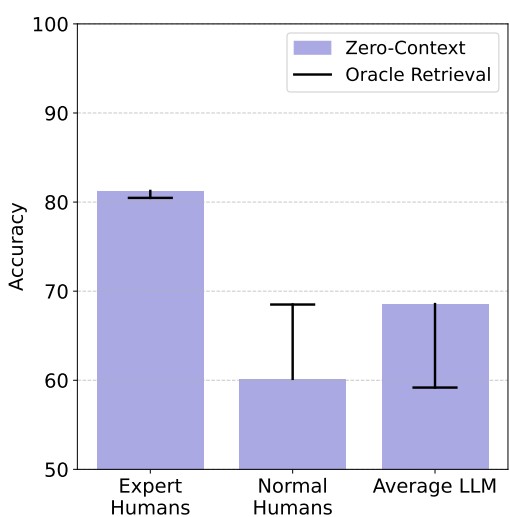

Shown in Figure 5, while normal humans benefit from having more information from a potentially misleading context, LLMs are easily misled by same information, illustrated by the longer black bar. In contrast, experts' consistently high accuracy suggests they do not benefit from additional information but can robustly handle misleading information. Rather than evaluating which of the three groups achieves higher accuracy on the zero-shot task, we focus on how they react to noisy context (i.e., the black bar in Figure 5). These results highlight a significant gap between human and machine reasoning.

Figure 5: Performance of humans and LLMs on a subset of RAGUARD, with black bars showing the performance differences when given documents.

## 6    Limitations

While the approach of using GPT-4 for document annotation has a model-specific element, akin to how human annotation carries annotator perspectives, its utility lies in creating a testbed for a unique aspect of LLM robustness. We assessed bias through a cross-model agreement analysis, which found moderate to strong consistency across LLM annotators, and confirmed in Section 4 that all models, not simply GPT-4, fail similarly on the same examples. We further emphasize that our task is fact-checking based on gold verdicts from PolitiFact rather than misinformation detection. GPT-derived labels are used only for intermediate analysis (i.e., determination of which documents are included in the Oracle Retrieval (Misleading) setting).

The limited scope of the human study constrains the strength of human–model comparisons, though it serves as a diagnostic check showing that observed failures arise from model reasoning rather than inherently impossible tasks. Ultimately, our main contribution lies in demonstrating how our

dataset harms and misleads LLMs, as it is specifically constructed to challenge model reasoning. Whether these instances also mislead humans is not the primary focus. Scaling human evaluation for fact-checking remains an open problem that future work should address.

## 7 Conclusion

In this paper, we highlight the importance of assessing the robustness of RAG systems against misleading retrievals, defining a new robustness-focused fact verification task that challenges models to reason through misleading content while unifying inconsistent terminology in prior works into a structured framework. To advance the development of robust fact-checking systems, we introduce RAGUARD, a diverse benchmark incorporating naturally occurring misleading data from Reddit discussions alongside verified evidence and claims from PolitiFact. Unlike prior RAG benchmarks that rely on synthetically noisy data, RAGUARD utilizes real-world evidence, targeting cases where gold-standard documents may not exist. This mirrors the complexities of real-world misinformation, which is necessary for more robust systems.

Our findings show that the performance of current RAG systems deteriorates significantly when exposed to misleading evidence, challenging the assumption that retrieval always enhances model accuracy. Consequently, future research should focus on enhancing LLM robustness through methods such as adversarial retrieval training, which exposes models to misleading evidence during training to improve resilience. Additionally, incorporating multi-step reasoning and classifying documents for their subjectiveness can mitigate the impact of misleading sources.

By providing a challenging yet realistic benchmark, RAGUARD encourages the development of more sophisticated retrieval-based fact-checking methods. We hope this dataset will facilitate progress in designing retrieval pipelines that are not only effective but also resistant to misinformation, ultimately contributing to more reliable and trustworthy AI systems.

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

# A Definitions

Prior works employ varying terminology to describe the presence of such noise in retrieved contexts or retrieval corpora [6, 22, 8, 38]. To establish consistency, we define a structured taxonomy and align existing definitions (See Figure 2).

Typical RAG datasets, including all prior fact-checking datasets to our knowledge, exclusively contain non-noisy, supporting documents as associated evidence, leading to overly optimistic performance [6]. Instead of relying solely on answer-containing documents, our dataset adopts a broader notion of supporting evidence. Specifically, we consider a document to be supporting if it provides information that enables an LLM to infer the correct answer, even if it does not explicitly state the ground-truth output. This reflects real-world fact-checking, where human verifiers rely on contextual information rather than single authoritative documents.

We categorize different types of noisy evidence based on whether the information directly conflicts with aspects of the correct prediction. As in prior work [6, 8], we include non-conflicting documents in RAGUARD, such as unrelated texts that may hurt performance. However, our primary focus is conflicting documents, which include misleading, fabricated, and unambiguous evidence. Previous datasets primarily include conflicting evidence as fabricated or unambiguous documents, oversimplifying real-world complexity and ambiguity (see Section 2.2 for further discussion) [38, 22, 8]. Notably, no prior work has introduced misleading documents.

In RAGUARD, misleading documents distort facts through selective framing, omission, or biased presentation, leading the system toward incorrect predictions while still containing partial truths. Unlike fabricated evidence, which is explicitly engineered to contradict the correct prediction (i.e., adversarial perturbations), misleading evidence subtly misguides the model rather than directly opposing it. Additionally, while prior work such as QACC [22] introduces unambiguous evidence—a term we adopt to ensure consistency with past research—which includes some naturally conflicting evidence but only for a limited set of unambiguous questions, we focus on more natural yet scalable conflicting evidence.

For reference, we provide a list of all defined terms. Each term defines a type of document or piece of evidence.

1. *Associated:* any document linked to a claim, regardless of label
2. *Supporting:* aids the system in producing a correct prediction through containing the correct answer explicitly or providing contextual support
3. *Noisy:* challenges or disrupt system performance, thereby enhancing robustness
4. *Conflicting:* contradicts either the correct answer or some aspect of the prediction
5. *Misleading:* introduces factual distortions through selective framing, omission, or biased presentation; may contain partial truths
6. *Fabricated:* synthetically constructed to include factual errors (e.g., adversarial perturbations)
7. *Unambiguous:* naturally conflicting evidence but only for a limited set of unambiguous questions (special case of [22])
8. *Non-Conflicting:* does not directly contradict the correct answer but still introduces noise by distracting the model
9. *Unrelated:* does not contain specific enough information to determine the correct prediction, despite being topically or semantically related to the query
10. *Random:* unrelated; often introduced through random selection or artificial generation

# B Dataset Statistics

Note that RAGUARD includes many unrelated documents, reflecting the natural distribution of the internet, where many discussions are neutral, neither misleading nor supporting the validity of a claim. These unrelated documents are topically relevant to their corresponding claim as they are retrieved with keyword search in the Dataset Construction stage (Section 3.2). However, they are labeled as

*unrelated* because they do not specifically provide misleading or supporting content (Section 3.3). Although less directly misleading, these documents still significantly reduce LLM accuracy (e.g., Llama 3 drops from 61.1% to 50.2%).

We provide additional dataset statistics in Figure 6. We see a spike in claims during recent presidential elections (2012, 2016, and 2024) which is an expected time for increased political discussion. We also note that more recent candidates usually have more claims in our dataset. This reflects the increasing popularity of fact-checking in recent years.

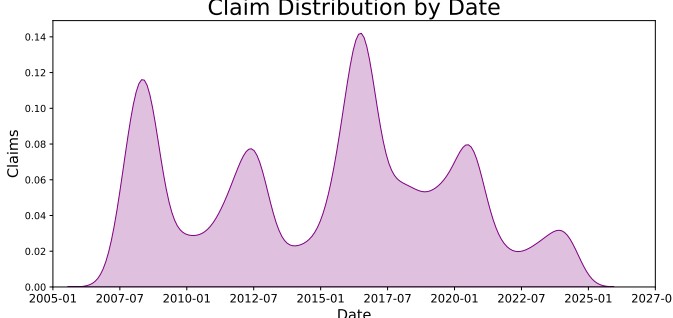

| Speaker | Claims |
|---|---|
| Al Gore | 6 |
| George W. Bush | 4 |
| John Kerry | 14 |
| John McCain | 173 |
| Barack Obama | 589 |
| Mitt Romney | 204 |
| Hillary Clinton | 295 |
| Donald Trump | 1008 |
| Joe Biden | 308 |
| Kamala Harris | 47 |

Figure 6: Claim distribution by date and speaker.

## C   Prompt Templates

For all settings and models, we used a temperature of 0.1.

### C.1   Dataset Creation

> **Keyword Extraction Prompt**
>
> Please take the following political claim and convert it into a set of a few (3-5) keywords that are optimized for searching Google.
> The keywords will be used to look for relevant Reddit articles. Make sure that the keywords together carry the idea of what the claim is saying so that you can find relevant posts for the claim on Google. The keywords should retain the meaning of the claim, but in a more concise form.
> Claim: ...
> Only provide the keywords, without any other text.

> **Document Labeling Prompt**
>
> You are an assistant that assesses the agreement between a DOCUMENT and a CLAIM.
> CLAIM: ...
> DOCUMENT: ...
> Based on the DOCUMENT, do you agree with the CLAIM? Respond with your reasoning and then with either AGREE or DISAGREE.
> If the DOCUMENT (even indirectly) supports the given CLAIM, respond with "AGREE".
> If the document (even indirectly) contradicts the CLAIM, respond with "DISAGREE". If there is no support (indirect or direct) against or for the claim, output unrelated.

### C.2   Evaluation

In all settings, two-shot examples—one true and one false claim from RAGUARD training data—are provided in the context.

You are an expert fact-checker. Given a claim and some context, your task is to assess its accuracy and classify it as either TRUE or FALSE.
Examples:
Claim: 'Obamacare' puts the federal government between you and your doctor.
Output: FALSE
Claim: $500 billion of the extension of the Bush tax cuts for the wealthy goes to 120,000 families.
Output: TRUE
Respond only with the following format, nothing else:
Output: [TRUE or FALSE]
Do not include any additional text, headers, or explanations outside this format.
Claim: ...

You are an expert fact-checker. Given a claim and some context, your task is to assess its accuracy and classify it as either TRUE or FALSE. The context is NOT ALWAYS relevant or correct. You must use your own judgement and must return a response.
Examples:
Claim: 'Obamacare' puts the federal government between you and your doctor.
Output: FALSE
Claim: $500 billion of the extension of the Bush tax cuts for the wealthy goes to 120,000 families.
Output: TRUE
Respond only with the following format, nothing else:
Output: [TRUE or FALSE]
Do not include any additional text, headers, or explanations outside this format.
Context: ...
Claim: ...

# D  Expanded Results and Analyses

Table 3 is a version of the results table (Table2) with exact relative percent decreases shown.

| | Open Source | | Mistral | Closed Source | | | Reasoning | |
|---|---|---|---|---|---|---|---|---|
| | OLMo-1B | Llama 3 | Mistral | Gemini 1.5 | GPT-4o | Claude 3.5 | DeepSeek | o4-mini |
| Zero-Context Prediction | 56.87 | 62.50 | 63.97 | 61.06 | 67.33 | 74.51 | 69.98 | 63.67 |
| RAG-1 | 52.68$_{\downarrow 7.4\%}$ | 59.40$_{\downarrow 5.0\%}$ | 59.14$_{\downarrow 7.6\%}$ | 56.68$_{\downarrow 7.2\%}$ | 64.80$_{\downarrow 3.8\%}$ | 70.09$_{\downarrow 5.9\%}$ | 66.88$_{\downarrow 4.4\%}$ | 62.76$_{\downarrow 1.4\%}$ |
| RAG-5 | 49.74$_{\downarrow 12.5\%}$ | 61.37$_{\downarrow 1.8\%}$ | 58.91$_{\downarrow 7.9\%}$ | 57.59$_{\downarrow 5.7\%}$ | 65.90$_{\downarrow 2.1\%}$ | 68.58$_{\downarrow 8.0\%}$ | 57.81$_{\downarrow 17.4\%}$ | 63.14$_{\downarrow 0.8\%}$ |
| Oracle Retrieval (All) | 53.89$_{\downarrow 5.2\%}$ | 61.09$_{\downarrow 2.3\%}$ | 51.55$_{\downarrow 19.4\%}$ | 52.38$_{\downarrow 14.2\%}$ | 53.22$_{\downarrow 21.0\%}$ | 52.56$_{\downarrow 29.5\%}$ | 50.06$_{\downarrow 28.5\%}$ | 51.88$_{\downarrow 18.5\%}$ |
| Oracle Retrieval (Misleading) | 44.04$_{\downarrow 22.6\%}$ | 36.81$_{\downarrow 41.1\%}$ | 26.88$_{\downarrow 58.0\%}$ | 30.57$_{\downarrow 49.9\%}$ | 45.97$_{\downarrow 31.7\%}$ | 35.98$_{\downarrow 51.7\%}$ | 38.25$_{\downarrow 45.3\%}$ | 33.39$_{\downarrow 47.6\%}$ |

Table 3: Accuracy (%) of various LLM backbones in RAG setup across three tasks and five evaluation settings. Subscripts indicate percent decrease from the Zero-Context baseline.

## D.1  GPT-4's Role in Annotation vs. Evaluation

Although GPT-4 was used during data construction to guide labeling, it does not necessarily have an inherent advantage on our benchmark. When used to label documents, GPT-4 does not directly classify documents as misleading or not. Instead, it produces a "test" prediction for each claim (i.e., *true* or *false*) given the retrieved documents, which is then compared against the dataset gold verdicts to derive dataset labels (see Section 3.3). Consequently, GPT-4 can still be misled by the same evidence when re-evaluated in the benchmark setting. GPT-4 does not necessarily know when evidence is misleading or not.

Because the dataset is partially constructed around GPT-4's own failures, GPT-4 is theoretically expected to perform worse on these misleading instances because they exploit its prior weaknesses. However, GPT-4 does not drastically underperform compared to models like DeepSeek. This suggests that the misleading patterns in the dataset are not uniquely adversarial to GPT-4 but rather reflect broader challenges that also affect other models.

We hypothesize that the fact that GPT-4 does not drastically underperform other models is because of its inherent strength as a model. While parts of the dataset may be harder for GPT-4 due to its role in data construction, GPT-4 also remains a very strong model at fact-checking relative to other models tested. Recent fact-checking papers benchmarking LLMs find GPT-4 to improve the best of the LLMs tested [30]. This reasoning strength likely offsets the disadvantage, leading to competitive performance despite the dataset being partially shaped by its own failure cases.

## D.2 Retrieval Performance

Retrieval performance is a standard metric in RAG benchmarks, but our dataset focuses on how models handle misleading or conflicting evidence. High retrieval accuracy alone does not ensure reliable answers due to misleading information in the corpus. Nonetheless, to provide a full view of system behavior, we report both conventional retrieval metrics and a tailored measurement called Misleading Retrieval Recall.

Figure 7 shows Retrieval Precision, Recall, and Normalized Discounted Cumulative Gain (NDCG) for Task 2 (Standard RAG). Recall naturally rises with $K$, the number of documents that are returned as the highest-ranked results, while precision decreases. NDCG follows a non-monotonic trend, dipping around $K = 10$ before recovering due to relevant items being unevenly distributed across ranked positions, causing re-ordering as $K$ changes.

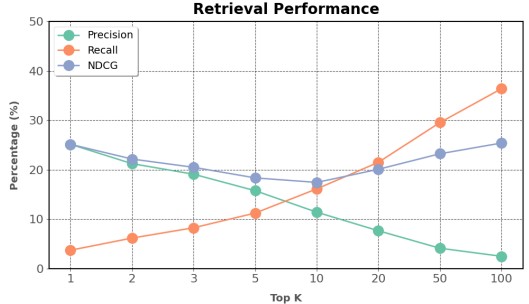

Figure 7: Retrieval Accuracy, Recall, and NDCG at Different Top K Levels

We also report Misleading Retrieval Recall—the fraction of claims retrieving at least one misleading document. Zero-Context Baseline scores 0%, while Oracle Retrieval scores 100%. RAG-1 scores 21.3%, increasing to 44.8% for RAG-5, showing a higher risk of retrieving misleading content when retrieving more documents. As seen in Table 2, this correlates with lower overall accuracy.

# E  Qualitative Examples and Error Analysis

## E.1  Failure Analysis

Our analysis reveals two primary failure modes that explain how misleading evidence interferes with LLM reasoning. First, models often misinterpret subjective language or rhetorical tone as factual support. For example, a user's frustrated question, "Is it normal to be taxed this much?" is incorrectly used to verify a claim about a tax hike.

Second, models frequently latch onto superficial details such as numbers or names while overlooking broader context. In one case, a document referencing a "300 million dollar" cost unrelated to the target claim was used to support a "300 billion dollar" fiscal figure. These examples demonstrate that LLMs overweight surface-level signals and struggle with nuanced reasoning tasks like assessing temporal context or implicit framing, limitations that explain their vulnerability to misleading retrieval.

## E.2  Types of Misleading Documents

Figure 8 presents example system predictions on RAGUARD, illustrating the impact of misleading documents. The left example highlights how misleading documents negatively affect the classification

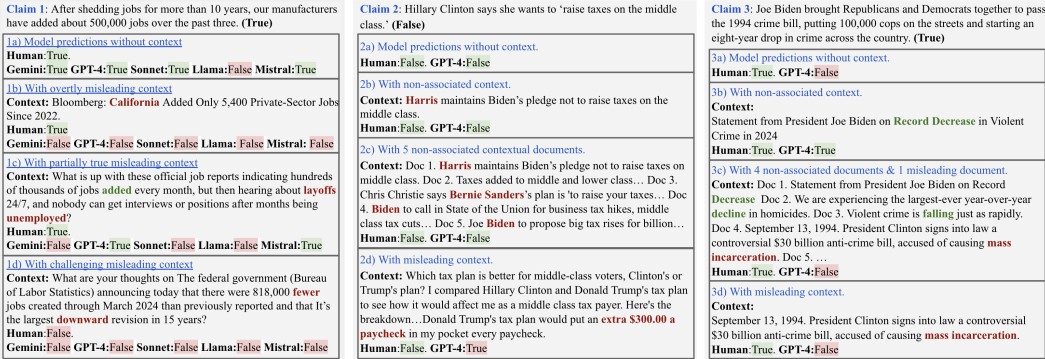

Figure 8: Example predictions on RAGUARD, compared to the expected human response. Note that each column compares different prediction scenarios based on varying retrieved contexts for the same claim rather than a multi-turn process. *Left:* Each system's classification of a true claim with three progressively misleading documents. *Middle:* GPT-4o-based system's classification of a false claim with one noisy non-associated document, many noisy non-associated documents, and a misleading document. *Right:* GPT-4o-based system's classification of a true claim with a supporting non-associated document, one misleading document along with other supporting non-associated documents, and a misleading document.

of a true claim. While misleading documents generally degrade system performance compared to zero-shot predictions, their specific influence varies based on their complexity. We distinguish three categories of misleading documents:

1. Overtly Misleading Document: This category includes documents that are evidently misleading to humans but still lead to incorrect predictions by all RAG systems. For example, in Figure 8, the document falsely comparing California's job growth to the national average misleads all systems (1b), despite their correct zero-shot predictions (1a). This suggests a form of selective bias, where the systems prioritize the provided information simply because it is included in the prompt, even though the instructions explicitly caution against assuming its correctness.

2. Partially True Misleading Document: These documents contain partial truths, making it necessary to apply reasoning to recognize their misleading nature. For example, as shown in Figure 8, one document criticizes unemployment but also states that "official job reports are reporting jobs added" (1c). While this statement supports the claim that 500,000 jobs were added, the document's overall tone suggests rising unemployment. However, this suggestion is more of an opinion than a fact. Some LLMs, such as GPT-4o and Mistral, were able to reason through this contradiction and classify the claim correctly.

3. Challenging Misleading Document: These documents present significant challenges, even for human annotators. For example, a claim referencing job growth in the 2000s is incorrectly classified because the RAG system retrieves data from 2024, which accurately reports lower job creation (1d). The temporal misalignment in retrieved documents presents a fundamental challenge in this dataset and task.

The middle example demonstrates GPT-4o's ability to filter out noise from retrieved documents that are not associated with the claim but could be considered misleading documents in our dataset (e.g., documents using the same phrasing but referring to different individuals, such as "Harris" instead of "Clinton" in example 2b). Even when five unrelated documents are retrieved (2c), GPT-4o remains robust. However, when presented with a misleading document from the dataset (2d), GPT-4o fails, reinforcing the dataset's effectiveness in challenging model performance beyond conventional RAG noise. This further explains the lower accuracy observed in the Oracle Retrieval setting in our baseline experiments.

The right example shows how GPT-4o tends to assign disproportionate weight to misleading documents, allowing them to override even non-associated supporting evidence. In the example, a non-associated document that contains supporting information (3b) enables GPT-4o to correct its

initially incorrect zero-shot prediction (3a). However, when a misleading document is retrieved alongside other non-associated supporting documents (3c), the system incorrectly classifies the claim, similar to its behavior when only the misleading document is retrieved (3d). This demonstrates that misleading documents can have a stronger influence on the model's classification, regardless of the presence of supporting evidence, highlighting a significant vulnerability in RAG systems.

These examples highlight three key findings. First, LLMs remain highly susceptible to misleading documents, even when their content is transparently incorrect. Second, misleading documents retrieved from the dataset exert a stronger influence than non-associated documents retrieved erroneously. Third, when misleading documents are present, they can significantly outweigh supporting evidence, leading to incorrect predictions. These findings emphasize the strength and uniqueness of our dataset in evaluating and challenging RAG-based model performance.

## F    Human Study Setup

We recruit four undergraduate and PhD students familiar with the U.S. political landscape. Annotators first make zero-context predictions, then are given one document without its label, and make predictions on the same claims, simulating the Oracle Retrieval setting. The annotators come from diverse backgrounds, both international and Californian, to reflect a range of political awareness. Figure 9 displays instructions, which are provided along with a spreadsheet to complete the predictions.

We construct a 64-instance subset (balanced across true and false claims) representative of the main contributions over our full dataset, with 20 misleading, 36 supporting, and 8 unrelated documents. While the misleading and supporting document distribution reflecting the distribution of misleading and supporting documents in the full dataset, we down-sample unrelated documents, which are less impactful to model performance.

**Participants Instructions**

You are tasked with evaluating the accuracy of a claim, aka., Fact-Checking. For each claim, you will follow two rounds:

You may start with the first sub-sheet in the file

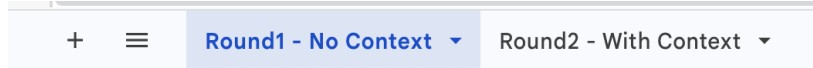

**Round 1: Evaluation Without Context**

1. Read the claim carefully.

2. Based only on your general knowledge, classify the claim as either **TRUE** or **FALSE** using the first column, i.e., Fact-Checking column.

    a. **TRUE:** If the claim is factually correct or mostly correct based on your knowledge.

    b. **FALSE:** If the claim is factually incorrect or misleading based on your knowledge.

3. Do not consider any context beyond your background knowledge at this stage.

Note that we believe you may feel unsure or clueless about many claims because they focus too much on details. That's okay, just give your best guess.

**Round 2: Evaluation With Context**

1. Review the provided context, which gives additional information.

    a. The Document Title column shows the title to the post; while the Document Text column includes the main text inside of the post.

    b. **Note:** The context may not always be fully relevant or correct, so make your assessment carefully. Your fact checking verdict may not always change after a given context.

        i. Specifically, the given context could be Support, Misleading, or Irrelevant (only a very small number irrelevant - less than 5).

2. Assess the claim's truthfulness again after reading the context.

    a. **TRUE:** If the claim is factually correct or mostly correct based on the context and/or your general knowledge.

    b. **FALSE:** If the claim is factually incorrect based on the context and/or your general knowledge.

**Important Note:** If the provided Document Text is a [Link Post], you should evaluate the claim based solely on the Document Title and your understanding, as no further context is available.

Figure 9: Instructions provided to humans to verify a subset of the dataset.

