# OpenReview forum: "Worse than Zero-shot? A Fact-Checking Dataset for Evaluating the Robustness of RAG Against Misleading Retrievals"
_NeurIPS.cc/2025/Datasets_and_Benchmarks_Track — NeurIPS 2025 Datasets and Benchmarks Track poster_

### Official Review · Reviewer_Ysze · 2025-06-22

**Rating:** 4
**Confidence:** 4

**Summary:**

This paper introduces RAGUARD, a novel benchmark dataset designed to evaluate the robustness of Retrieval-Augmented Generation (RAG) systems against misleading retrievals. Unlike prior benchmarks that assume clean retrieval settings, RAGUARD incorporates real-world misinformation from Reddit discussions, categorizing evidence as supporting, misleading, or unrelated. They find that current RAG systems perform worse than zero-shot baselines when exposed to misleading retrievals, highlighting a critical vulnerability in LLM-based fact-checking.

**Dataset Code Accessibility:**

Yes

**Dataset Code Comments:**

The dataset link is provided.

**Ethical Considerations:**

No, there are no or only very minor ethics concerns

**Final Justification:**

Appreciate your response. There are no additional concerns on my part.

**Limitations Weaknesses:**

While the paper mentions potential misuse (e.g., political misinformation), it could further discuss mitigation strategies

The dataset contains a high proportion of unrelated documents (72.5%), which may dilute the challenge posed by misleading evidence. A more balanced distribution is required.

Since GPT-4 was used to collect data and determine whether content aligns with facts, and then re-evaluate it, theoretically GPT-4 should perform exceptionally well. However, why didn't GPT-4 significantly outperform other models like DeepSeek in the results?

The paper lacks comparison with existing robustness methods (e.g., retrieval filtering, confidence thresholding, multi-step reasoning approaches

**Strengths Contributions:**

- The paper addresses the gap in RAG evaluation by focusing on robustness against misleading retrievals, a realistic but understudied challenge.
- The paper provides a **well-defined taxonomy** of document types (supporting, misleading, unrelated) and aligns terminology with prior work.

---

> ### Author Rebuttal · Authors · 2025-07-31
>
> We thank Reviewer Ysze for the feedback and for acknowledging our well-defined taxonomy of document types and how our paper aligns terminology with prior work. We appreciate the acknowledgement of our paper’s contribution in addressing the important and understudied challenge of RAG robustness against misleading retrievals. We address the reviewer’s concerns below with new experiments and analysis.
>
> ***1. Concerns on Dataset Misuse and Safety***
>
> We appreciate the reviewer’s concern regarding potential misuse, as this is indeed an important issue in real-world applications. However, we believe this concern falls *outside the scope* of our research focus. Our dataset is built from two sources: Politifact and Reddit. The Politifact portion consists of fact-checked claims. The Reddit portion, similar to many other Reddit-based datasets, inherits any safety considerations from the platform itself. We do not introduce synthetic misinformation or alter the original content, and therefore do not amplify risk beyond what is *already present* in the source material.
>
> Furthermore, our dataset is released with a warning card and includes links to all original Reddit documents to ensure proper attribution and transparency. As noted in lines 101–104, the Reddit documents primarily capture subjective user perspectives rather than deliberate misinformation. While isolated cases of user-spread misinformation may exist, they are *neither fabricated nor injected* by our pipeline. Consequently, we do not view this as a limitation of our dataset but rather an inherent property of using real-world, user-generated content—one we have explicitly acknowledged, provided clear warnings for, and mitigated through measures such as sourcing original URLs in our release.
>
> ***2. "Unrelated" Documents***
>
> The reviewer raises an interesting point about the dataset's distribution. We argue that the high proportion of "unrelated" documents is a feature that reflects real-world retrieval and exposes a distinct type of model vulnerability.
>
> * **Reflecting Reality:** In practice, search queries often return documents that are topically related but do not directly contain the answer. Our distribution faithfully captures this "needle in a haystack" problem.
>
> * **"Unrelated" is Still a Challenge:** These documents are not benign. To demonstrate this, we ran a **new experiment** evaluating models on *only* the unrelated documents. The results show a significant performance drop compared to the zero-shot baseline, confirming that even topically related but uninformative documents can confuse current LLMs.
>
> | Model                | Llama 3 | Mistral | Gemini 1.5 | GPT-4o | Claude 3.5 | DeepSeek | o4-mini |
> |----------------------|---------|---------|------------|--------|------------|----------|---------|
> | **Zero-Context Prediction**    | 62.50   | 63.97   | 61.06      | 67.33  | 74.51      | 69.98    | 63.67   |
> | **Oracle Retrieval (All)**     | 61.09   | 51.55   | 52.38      | 53.22  | 52.56      | 50.06    | 51.88   |
> | **Oracle Retrieval (Irrelevant)** | 59.90   | 48.59   | 50.32      | 52.84  | 51.58      | 49.50    | 50.25   |
> | **Oracle Retrieval (Misleading)**| 36.81   | 26.88   | 30.57      | 45.97  | 35.98      | 38.25    | 33.39   |
>
> * **Dedicated Subtask for Misleading Evidence:** To ensure the challenge of misleading evidence is not diluted, our evaluation already includes a dedicated **Oracle Retrieval (Misleading)** subtask (lines 197-199). This isolates the most challenging cases and evaluates models specifically on their ability to handle deceptive content, a point that remains central to our contribution.
>
> ***3. Clarifying GPT-4’s Role in Annotation vs. Evaluation Performance***
>
> Although GPT-4 was used during data construction to guide labeling, it does not necessarily have an inherent advantage on our benchmark. There are several reasons for this:
>
>
> 1. The task is not misinformation detection. In Dataset Construction, GPT-4 is *not* classifying documents as misleading or not. Instead, it produces a “test” prediction for each claim (i.e. True or False) given the retrieved documents, which is then compared against the dataset gold verdicts to derive dataset labels. As described in line 228 and on, “We annotate each retrieved document based on its **functional effect on the model’s prediction**…” This means GPT-4 can still be misled by the same evidence when re-evaluated in the benchmark setting. GPT-4 does *not* necessarily *know* when evidence is misleading or not.
>
>
> 2. As elaborated in the paragraph above, the dataset is partially constructed around GPT-4’s own failures. Therefore, GPT-4 is theoretically expected to **perform worse** on these misleading instances because they exploit its prior weaknesses. This also allows us to test whether other models are misled by the same patterns.
>
>
> 3. Observed performance suggests cross-model vulnerability. Interestingly, GPT-4 does not drastically underperform compared to models like DeepSeek. This suggests that the misleading patterns in the dataset are not uniquely adversarial to GPT-4 but rather reflect **broader challenges** that also affect other models.
>
>
> 4. GPT-4’s inherent strength may counterbalance dataset bias. We agree that it may seem like GPT-4 should still perform **differently** compared to other models. We hypothesize that the fact that this does not occur is that, while parts of the dataset may be harder for GPT-4 due to its role in data construction, GPT-4 also remains a very strong model at fact-checking relative to other models tested. Recent fact-checking papers benchmarking LLMs like OpenFactCheck (Wang et al., 2025) and MiniCheck (Tang et al., 2025) find GPT-4 to improve the best of the LLMs tested. This reasoning strength likely offsets the disadvantage, leading to competitive performance despite the dataset being partially shaped by its own failure cases.
>
>
> In summary, although GPT-4 contributed to data construction, it does not gain an inherent advantage because the benchmark is built around its own points of failure and does not involve detecting misleading evidence directly. While GPT-4 is expected to struggle on misleading cases, the results show that these challenges affect all models, not just GPT-4. Its strong fact-checking and reasoning capabilities (Wang et al., 2025; Tang et al., 2025) likely counterbalance its role in annotation, resulting in performance that is competitive—but not substantially better—than other strong models like DeepSeek.
>
>
> ***4. Comparison with Existing Robustness Method***
>
> We appreciate the reviewer’s suggestion to evaluate existing advanced robustness methods under our setting. To this end, we implemented **Corrective RAG (CRAG)** (Yan et al., 2024), a method suggested by the reviewer mwAZ.
>
> * **Corrective RAG:** CRAG enhances retrieval quality through a lightweight evaluator that scores document relevance, triggers alternative retrieval strategies, and applies a decompose–recompose algorithm to filter key information. When evidence from the static corpus is judged insufficient, CRAG supplements it with large-scale web searches.
>
>
> * **Implementation:** Using the authors’ released code, prompts, hyperparameters, and trained Critic parameters, we conducted *new* experiments and ran inference on our dataset using our documents as retrieval input (Oracle Retrieval setting) with both their Llama-2 and SelfRAG-Llama-2 as generators. For comparison, we also evaluated Llama-2 in a zero-shot setting. The accuracy is reported:
>
>
>   * Zero-Shot (Llama-2): **50.57%**
>
>
>   * CRAG (Llama-2): **37.24%**
>
>
>   * CRAG (SelfRAG-Llama-2): **35.55%**
>
>
> * **Low Results:** Despite being designed to mitigate retrieval errors, CRAG performs substantially *worse* than the zero-shot baseline, showing a 26% percent decrease with Llama-2. We find that its evaluator indeed triggers web search when documents are deemed low-quality (70.1% of the time), yet the retrieved web content often adds further noise or misleading evidence. In RAGuard, claims are complex, real-world, and rarely verifiable by straightforward web sources such as Wikipedia, suggesting why this method may not yield high performance. Furthermore, when the evaluator’s confidence is low, it combines web search results with oracle documents, effectively doubling exposure to misleading content and compounding errors.
>
>
> * **Gap in Current Robustness Method:** Therefore, we believe that CRAG does sense the challenge of the problem, but it is still unable to resist the misleading information in the retrieval and is finally misled. More broadly, this underscores that existing robustness methods, which have been shown to handle certain type of noise, remain vulnerable to the qualitatively different challenge of real-world misleading retrieval.

---

### Official Review · Reviewer_xwdU · 2025-06-24

**Rating:** 5
**Confidence:** 3

**Summary:**

This paper introduces a benchmark to examine the robustness of RAG in political fact-checking. It pairs laims from PolitiFact with Reddit. Experiments the vulnerability of RAG.

**Dataset Code Accessibility:**

Yes

**Ethical Considerations:**

No, there are no or only very minor ethics concerns

**Final Justification:**

The authors provided additional results during the rebuttal and I have no additional concerns.

**Limitations Weaknesses:**

[1] The main concern is how the data are being annotated. The authors leverage GPT-4 to annotate the data to label it as supporting, misleading, or unrelated (to see whether GPT-4 's answer is correct, wrong, or unused). However, it is not clear whether the choice of GPT-4 will impact the data quality or not. Since different LLM models have different reasoning abilities (e.g., comparing with GPT-4o or O1) and the pre-training knowledge are also different (e.g., comparing with Gemini or Claude, or newer/older models), the authors are encouraged to try other LLMs and compare the interrate agreement.

[2] The authors may also try different retrievers. Currently the data collection relies on text-embedding-ada-002. Methods such as text-embedding-3-large or edit distance may also be considered.

[3] The benchmark only includes samples related to US (and in English). It is not clear how RAG performs in other languages. All the data quality (Reddit may have more noisy data for other country's posts), LLM model (e.g., LLMs from other countries), and retriever can affect this.


In addition, the writing of this paper can also be improved. For example,

line 59: that that

line 190: to retrieved

Figure 4: the each

line 264: predicton

line 325: by same information

**Strengths Contributions:**

[1] The benchmark uses Reddit discussions rather than synthetic edits. This can reflect the real-world scenario better.

[2] The human study result is interesting.

---

> ### Author Rebuttal · Authors · 2025-07-31
>
> We thank Reviewer xwdU for the constructive feedback and for recognizing the value of using real-world Reddit discussions in our benchmark. We have conducted new experiments and analysis to address the reviewer's excellent points.
>
>
> ***1. Quality of GPT-4 Labels***
>
> We thank the reviewer for their suggestion to conduct an inter-annotator agreement study on different LLMs as annotators to evaluate the potential impact of using GPT-4 annotation on data quality.
>
> We report agreement on `supporting` and `misleading` labels, the two classes that are derived from the LLM’s predictions. The distinction between these two determines whether retrieved evidence led the model toward a correct or incorrect answer during the annotation stage, whereas `unrelated` documents do not contribute to a model’s prediction during the data annotation stage. Specifically, we had Claude 3.5 Sonnet and Gemini 1.5 Flash re-annotate the documents and found substantial (0.789 Kappa) and moderate (0.650 Kappa) agreement with GPT-4's labels, respectively.
>
> **Furthermore, we emphasize that the primary task is fact-checking against PolitiFact verdicts.** The GPT-4-derived labels are an analytical tool, and the benchmark is explicitly designed to be built around a model's failure modes to test if other models share the same vulnerabilities.
>
> ***2. Different Retrievers***
>
> We thank the reviewer for this insightful suggestion. We would like to clarify a point about our methodology and then address the suggestion directly.
>
> * **Dataset Collection vs. Baseline Retrieval:** Our dataset was collected using a keyword-based Google Search on Reddit (Section 3.2), not `text-embedding-ada-002`. The ada-002 model was used only for the *Standard RAG baseline* in our experiments (Section 4.1).
>
> * **Why a Different Retriever Won't Change the Core Finding:** Our experiments already include an **Oracle Retrieval** setting. This setting simulates a perfect retriever by directly providing the model with the ground-truth associated documents for a claim, completely removing any noise or errors from the `ada-002` retriever. Our key finding is that performance drops *even more* in this Oracle setting than in the Standard RAG setting.
> This demonstrates that the performance bottleneck is not the quality of the retriever, but the **inherently challenging nature of the documents themselves**. Even with a perfect retriever, the models are misled. Therefore, while a different retriever like `text-embedding-3-large` might alter the specific documents retrieved in the Standard RAG setting, it would not change our main conclusion, which is robustly supported by the Oracle setting results. Acknowledging the reviewer's point, we agree that a full-scale evaluation of different retrievers is a valuable direction for future work and have added a note to this effect in our limitations section.
>
> ***3. Multiple Languages***
>
> We thank the reviewer for this insightful comment. We agree that evaluating RAG robustness in multilingual and global contexts is an important avenue for future research. Our current benchmark focuses on U.S. political claims to ensure controlled annotation grounded in PolitiFact verdicts and to leverage document retrieval for claims that are widely discussed on the internet.
>
> **Importantly, our data construction pipeline (Figure 4)—which pairs real-world claims with retrieved user-generated documents and annotates misleadingness via LLM behavior—can naturally generalize to other regions and languages.** For example, future work could build on multilingual fact-checking datasets such as X-Fact (Vaidya et al., 2021) to source claims in non-English languages.

---

> > ### Comment · Reviewer_xwdU · 2025-08-03
> >
> > I appreciate the authors in the response and providing additional experiment results. Given the two kappa numbers, I would suggest the authors keeping at least either Claude or Gemini generated data in the dataset as a supplement. I have read the review of other reviewers, and would like to raise my score from 4 to 5. The approach in this paper and the insights are interesting to me.

---

> > > ### Author Response · Authors · 2025-08-04
> > >
> > > Thank you very much for reviewing our paper and recognizing our work. We are glad that our clarification addresses your concerns.
> > >
> > > We will include the discussed points in the final version.

---

### Official Review · Reviewer_mwAZ · 2025-06-26

**Rating:** 4
**Confidence:** 4

**Summary:**

The paper points out that RAG systems lack robustness when faced with misleading retrievals. To address this issue, the authors propose the RAGUARD dataset, which is built on PolitiFact political statements and Reddit discussions, containing three categories of evidence: supportive, misleading, and irrelevant. The goal is to evaluate the performance of RAG systems under real misleading scenarios. Experiments show that all tested LLM-driven RAG systems perform worse than the zero-shot baseline when encountering misleading retrievals, while human annotators outperform them, highlighting the vulnerability of LLMs in noisy environments.

**Additional Feedback:**

As highlighted in Strengths Contributions and Weakness, I highly appreciate the authors' exploration in this direction, which is valuable for the community. However, at this point, there are still certain deficiencies in the scientific integrity and completeness of the dataset design, the adequacy of the experiments, and the depth of the analysis.

**Dataset Code Accessibility:**

Yes

**Dataset Code Comments:**

The dataset is complete and publicly accessible.

**Ethical Considerations:**

No, there are no or only very minor ethics concerns

**Final Justification:**

This submission targets the important issue of RAG system robustness by introducing a new benchmark dataset featuring misleading retrievals in the political domain. The problem setting and dataset design are appropriate and respond to a clear need in the community.

The authors have addressed several key concerns in their rebuttal:

- They clarified the distinction between misleading and general retrieval noise, and updated claims about the novelty accordingly.
- Additional experiments with advanced robustness baselines (e.g., CRAG) were included, confirming that current methods remain highly vulnerable to the proposed dataset.
- Annotation consistency across multiple LLMs was measured, partially mitigating concerns over potential bias.

However, some limitations remain:

- The main contribution lies in dataset construction; novel technical or analytical insights are limited, and analysis of model failure modes remains preliminary.
- The human evaluation is underpowered due to its small scale, restricting the strength of related conclusions.

On balance, I find the work well-motivated and potentially impactful as a benchmark, but not fully convincing in analytical depth or novelty. I therefore recommend a borderline accept (score: 4).

**Limitations Weaknesses:**

1. Firstly, investigating the impact of noise introduced by retrievals in RAG is a widely researched topic. The paper's findings regarding LLM sensitivity to noise have been proven in many prior studies. However, the analysis and conclusions in this paper do not delve deeper than previous work.


2. In existing evaluation benchmarks, many have included this aspect of retrieval noise. Even in the early HotpotQA, noise segments were introduced through BM25. Recent work, such as CRAG, has been designed to include noisy segments through the SearchAPI format. It is incorrect to claim, as stated in the abstract, that all used "clean" retrieval settings.


3. The baseline is somewhat weak, especially since it only compares with standard RAG methods  across several LLMs while many RAG approaches have been proposed specifically to mitigate the interference of noise.


4. The comparison study between human and model performance has a rather small sample size, with only 2 experts and 2 ordinary users testing 64 statements. This makes it difficult to support the authors' conclusion that human annotators perform better.


5. Additionally, as mentioned by the authors in the limitations, using GPT-4 as the standard document type raises concerns. Relying too heavily on a single model can introduce biases and preferences inherent in the GPT series, potentially leading to skewed evaluation results of other models on this dataset, hindering accurate assessment of their ability to handle misleading information in real scenarios.


6. The analysis section is somewhat superficial; a more in-depth analysis is desired. How do the interference factors affect LLM decision-making? Is it more reliant on numerical values (as in Figure 1), or on textual descriptions? What typical error patterns exist, and how can they be mitigated?

**Strengths Contributions:**

1. The area and scenarios targeted by RAGUARD are valuable; there is currently a lack of evaluation benchmarks in the RAG field related to politics and policy. They typically have clear factuality while facing significant interference from outdated or false information, which can effectively support the advancement of related research.

2. The data comes from real-world scenarios, making it more difficult to reflect the complexities of real situations compared to synthetic data.

---

> ### Author Rebuttal · Authors · 2025-07-31
>
> We thank Reviewer mwAZ for the detailed feedback and for recognizing the value of our dataset's focus on the political domain. The reviewer raises several important points regarding the novelty of our work, the strength of our baselines, and the depth of our analysis, which we address below with new experiments and clarifications.
>
> ***Novelty of Studying RAG Noise***
>
> We agree with the reviewer that prior work has studied the impact of noise in RAG. Our contribution is not to rediscover that noise is harmful, but to introduce a new, more challenging, and realistic type of noise and a benchmark to measure it.
>
> * **Misleading vs. Generic Noise:** Our work focuses specifically on **misleading retrieval,** which is qualitatively different from the noise in prior benchmarks like HotpotQA (distractors) or CRAG (API noise). As detailed in our paper (lines 101-111), misleading documents in RAGuard are not simple factual errors but plausible, selectively framed, and opinion-laden texts from real-world Reddit discussions. This presents a unique challenge that tests a model's ability to reason through nuanced rhetoric, not just filter irrelevant text. We will refine the expression in the abstract to reflect this point.
>
> * **A New Domain and Taxonomy:** We are the first to systematically tackle this problem in the high-stakes political domain, where such misleading content is rampant. Our taxonomy (Figure 2) helps formalize the distinction between misleading content and other types of noise, a contribution another reviewer highlighted as a strength.
>
> * **Exposing Preference Alignment Failures:** Our findings also reveal a deeper issue: LLMs are overly deferential to any information provided in the context, even when cautioned. This highlights a failure in preference alignment, where models struggle to apply skepticism.
>
>
>
>
> ***3. Comparison with Advanced Robustness Baseline***
>
> We appreciate the reviewer’s suggestion to evaluate an existing advanced robustness method under our setting. To this end, we implemented **Corrective RAG (CRAG)** (Yan et al., 2024), which is the method suggested by the reviewer.
>
> * **Corrective RAG:** CRAG enhances retrieval quality through a lightweight evaluator that scores document relevance, triggers alternative retrieval strategies, and applies a decompose–recompose algorithm to filter key information. When evidence from the static corpus is judged insufficient, CRAG supplements it with large-scale web searches.
>
>
> * **Implementation:** Using the authors’ released code, prompts, hyperparameters, and trained Critic parameters, we conducted *new* experiments and ran inference on our dataset using our documents as retrieval input (Oracle Retrieval setting) with both their Llama-2 and SelfRAG-Llama-2 as generators. For comparison, we also evaluated Llama-2 in a zero-shot setting. The accuracy is reported:
>
>
>   * Zero-Shot (Llama-2): **50.57%**
>
>
>   * CRAG (Llama-2): **37.24%**
>
>
>   * CRAG (SelfRAG-Llama-2): **35.55%**
>
>
> * **Low Results:** Despite being designed to mitigate retrieval errors, CRAG performs substantially *worse* than the zero-shot baseline, showing a 26% percent decrease with Llama-2. We find that its evaluator indeed triggers web search when documents are deemed low-quality (70.1% of the time), yet the retrieved web content often adds further noise. In RAGuard, claims are complex, real-world, and rarely verifiable by straightforward web sources such as Wikipedia, suggesting why this method may not yield high performance. Furthermore, when the evaluator’s confidence is low, it combines web search results with oracle documents, effectively doubling exposure to misleading content and compounding errors.
>
>
> * **Gap in Current Robustness Method:** Therefore, we believe that CRAG does sense the challenge of the problem, but it is still unable to resist the misleading information in the retrieval and is finally misled. More broadly, this underscores that existing robustness methods, which have been shown to handle certain type of noise, remain vulnerable to the qualitatively different challenge of real-world misleading retrieval.
>
>
>
> ***4. Size of Human Study***
>
> We acknowledge the reviewer’s concern that the human evaluation uses only 2 experts and 2 ordinary users on 64 statements, and we agree that this limits the strength of claims about human–model comparisons. Scaling human evaluation for fact-checking remains an open problem that future work needs to address.
>
> Our main contribution lies in demonstrating how our dataset harms and misleads LLMs, as it is specifically constructed to challenge model reasoning. Whether these instances also mislead humans is not the primary focus. The small human study was intended only as a diagnostic check—to see whether the documents are inherently too difficult or whether the errors observed stem from model-specific alignment and bias issues, which would align with concerns raised in past work on noisy RAG.
>
> As affirmed by reviewer xwdU, our human study reveals interesting findings, e.g., these documents tend to mislead humans less and may even assist ordinary users in answering questions more effectively.
>
> ***5. Potential for Bias in GPT-4 Annotation***
>
>  We appreciate the reviewer’s concern about potential biases arising from using GPT-4 during annotation. While our experimental results remain consistent across models, we acknowledge the importance of validating label consistency with other LLMs. To address this, we tested two additional models, Claude and Gemini, for annotating documents and compared their document ratings with those of GPT-4 using Cohen’s Kappa Score.
>
> We report agreement on `supporting` and `misleading` labels, the two classes that are derived from the LLM’s predictions. The distinction between these two determines whether retrieved evidence led the model toward a correct or incorrect answer during the annotation stage, whereas `unrelated` documents do not contribute to a model’s prediction during the data annotation stage. Specifically, we had Claude 3.5 Sonnet and Gemini 1.5 Flash re-annotate the documents and found substantial (0.789 Kappa) and moderate (0.650 Kappa) agreement with GPT-4's labels, respectively.
>
> **Furthermore, we emphasize that the primary task is fact-checking against PolitiFact verdicts.** The GPT-4-derived labels are an analytical tool, and the benchmark is explicitly designed to be built around a model's failure modes to test if other models share the same vulnerabilities.
>
> ***6. On the Depth of Analysis***
>
> The reviewer asks for a deeper analysis of *how* interference affects LLMs. Our analysis shows two primary failure modes:
>
>  1. **Confusing Opinion with Fact:** Models often misinterpret the sentiment or tone of a Reddit post as factual evidence. For example, a user's frustrated question, "Is it normal to be taxed this much?", was used to incorrectly verify a claim about a tax hike.
>
>  2. **Misapplying Contextual Cues:** Models latch onto superficial cues like numbers or names while ignoring the broader context. For a claim about a "300 billion dollar" fiscal cost, a document mentioning a "300 million dollar" cost on a different topic caused an incorrect prediction.
>
> These patterns, detailed further in our revision, show that models overweight surface-level features and lack the deeper reasoning needed to evaluate the true relevance and veracity of evidence.

---

### Official Review · Reviewer_sXya · 2025-07-01

**Rating:** 5
**Confidence:** 4

**Summary:**

This paper introduces the RAGUARD dataset for systematically evaluating the robustness of RAG when faced with misleading search results. The authors use political fact-checking as a scenario, collecting real controversial statements from PolitiFact and scraping natural language posts related to the statements from Reddit. They use LLM to automatically label the impact of documents on model predictions, constructing a search-fact-checking benchmark with real noise. Through extensive experiments, the authors demonstrate that existing RAG systems perform even worse than zero-shot retrieval under misleading retrieval conditions, and provide human comparison experiments to highlight the susceptibility of LLMs to misleading information.

**Dataset Code Accessibility:**

Yes

**Ethical Considerations:**

No, there are no or only very minor ethics concerns

**Final Justification:**

This paper makes a valuable contribution by being the first to systematically study the robustness of RAG under misleading retrieval, addressing a critical gap overlooked in prior “idealized retrieval” evaluations. The analysis across seven mainstream LLMs, supported by human evaluation, provides strong evidence of current vulnerabilities. While I had concerns regarding the subjectivity of “misleading” labels, temporal variability, and potential GPT-4 bias, most of these were adequately addressed during the rebuttal. Overall, I find the contribution meaningful and maintain my accept score.

**Limitations Weaknesses:**

1.	Since “misleading” is defined relative to whether GPT-4 is misled, rather than as an objective property of the documents themselves, future improvements in model knowledge or reasoning could result in these documents are no longer misleading. This implies that the dataset labels may change over time. Have the authors considered any measures to address this potential temporal variability?

2.	The claim that “all tested systems perform worse with misleading retrieval than with none” is supported for the chosen models. I am curious whether smaller models or models with different parameter scales would exhibit similar confusion, and how different parameter versions of the same model might affect the results. This impacts how convincing the statement that “RAG can harm performance” is.

3.	Labels are based on GPT-4 judgments, which could carry its own biases. If GPT-4 is more easily misled by certain viewpoints, labels might lean toward specific political stances. PolitiFact helps balance this, but there’s no thorough check to ensure the labels are politically neutral.

4.	Figure 3(c) seems unclear or possibly corrupted.

**Strengths Contributions:**

1.	This work is the first to systematically address the robustness of RAG under misleading retrieval, highlighting the limitation of current RAG evaluations that rely on “idealized retrieval” (supporting documents only) and thus fail to reflect real-world scenarios with misleading or irrelevant content.

2.	A detailed analysis of seven mainstream LLMs (both open- and closed-source) across different retrieval scenarios was conducted, and human evaluations were used to verify their vulnerability.

---

> ### Author Rebuttal · Authors · 2025-07-31
>
> We thank Reviewer sXya for the positive assessment and insightful feedback. We are encouraged that the reviewer recognizes our work as the first to systematically address RAG robustness with a real-world dataset and appreciates our detailed analysis. We address the reviewer's specific concerns below.
>
> ***1. Temporal Variability of Labels***
>
> We agree with the reviewer's sharp observation: as models improve, documents currently labeled "misleading" may cease to be so, making the labels time-dependent. We see this not as a weakness, but as a fundamental and intended feature of a benchmark designed to push the boundaries of current model capabilities.
>
> * **Benchmarks Evolve:** Like ImageNet, which was effectively "solved" as models surpassed human performance, our benchmark is designed to highlight the current frontier of LLM reasoning. If future models consistently overcome the challenges in RAGuard, it will be a clear signal that the field has advanced—precisely the goal of our work.
>
> * **A Measure of Progress:** The temporal nature of our labels provides a dynamic measure of progress. The rate at which documents transition from "misleading" to "benign" can serve as a metric for evaluating genuine improvements in model robustness over time.
> Therefore, we argue this temporal aspect is a core strength, providing a durable tool for tracking the evolution of model resilience against real-world, nuanced misinformation.
>
> ***2. Generalizability to Smaller Models***
>
> The reviewer rightly questions whether the performance degradation we observe is unique to large models. To address this, we have conducted new experiments with **OLMo-1B,** a significantly smaller open-source model. The results, shown below, confirm that this vulnerability is not scale-dependent.
>
> | Setting                     | OLMo-1B (Open Source)   | Llama 3 (Open Source) | Mistral (Open Source) | Gemini 1.5 (Closed) | GPT-4o (Closed) | Claude 3.5 (Closed) | DeepSeek (Reasoning) | o4-mini (Reasoning) |
> |-----------------------------|--------------------------|------------------------|-----------------------|---------------------|-----------------|---------------------|----------------------|---------------------|
> | **Zero-Context Prediction** | 56.87                    | 62.50                  | 63.97                 | 61.06               | 67.33           | 74.51               | 69.98                | 63.67               |
> | **RAG-1**                   | 52.68 (-7.4%)            | 59.40 (-5.0%)          | 59.14 (-7.6%)         | 56.68 (-7.2%)       | 64.80 (-3.8%)   | 70.09 (-5.9%)       | 66.88 (-4.4%)        | 62.76 (-1.4%)       |
> | **RAG-5**                   | 49.74 (-12.5%)           | 61.37 (-1.8%)          | 58.91 (-7.9%)         | 57.59 (-5.7%)       | 65.90 (-2.1%)   | 68.58 (-7.9%)       | 57.81 (-17.4%)       | 63.14 (-0.8%)       |
> | **Oracle Retrieval (All)**  | 53.89 (-5.2%)            | 61.09 (-2.3%)          | 51.55 (-19.4%)        | 52.38 (-14.2%)      | 53.22 (-20.9%)  | 52.56 (-29.4%)      | 50.06 (-28.5%)       | 51.88 (-18.5%)      |
> | **Oracle Retrieval (Misleading)** | 44.04 (-22.6%)         | 36.81 (-41.1%)         | 26.88 (-58.0%)        | 30.57 (-49.9%)      | 45.97 (-31.7%)  | 35.98 (-51.7%)      | 38.25 (-45.3%)       | 33.39 (-47.6%)      |
>
> Our experiments show that all RAG methods using OLMo‑1B underperform compared to its zero-shot baseline, consistent with our findings on other models. Moreover, Oracle Retrieval (Misleading) produces the lowest accuracy across settings, reaffirming that misleading documents in our dataset significantly degrade performance. This confirms that the observed vulnerability is not limited to large models but also extends to smaller parameter scales. Interestingly, RAG-1 and RAG-5 appear to mislead OLMo‑1B more than they do with other models, while the Oracle Retrieval settings cause smaller relative decreases compared to other models. This is likely because OLMo‑1B starts from a lower zero-shot performance, making it more susceptible to confusion from any type of retrieved document, whether from RAG or Oracle Retrieval.
>
> ***3. Potential for Political Bias in GPT-4 Annotation***
>
> This is a critical point. We address it with three clarifications on our methodology:
>
> * **Task Framework:** The primary task is **fact-checking,** where ground-truth verdicts come from the reputable, human-driven organization PolitiFact. The "misleading" labels are an intermediate analytical tool to characterize document effects; they are not part of the final evaluation metric. The model's task is always to predict the PolitiFact verdict.
>
> * **Exposing Failure Modes is the Goal:** Our benchmark is designed to probe LLM behavior. If GPT-4's annotation process reveals a political bias in its reasoning (i.e., it is more easily misled by certain viewpoints), then our dataset has successfully captured a critical, measurable failure mode. This is a feature, not a bug, as the objective is to surface these vulnerabilities for the community to address.
>
> * **Cross-Model Validation:** To ensure the "misleading" patterns are not unique to GPT-4, we conducted a new inter-annotator agreement study. We had Claude 3.5 Haiku and Gemini 1.5 Flash re-annotate the documents. The Cohen's Kappa scores for agreement on `supporting` vs. `misleading` labels are:
>
>   * GPT-4 vs. Claude 3.5: **0.789 (Substantial Agreement)**
>
>   * GPT-4 vs. Gemini 1.5: **0.650 (Moderate Agreement)**
>
> This demonstrates that the phenomena captured by our labels are not a GPT-4 idiosyncrasy but reflect a broader challenge across leading LLMs.
>
> ***4. Clarity of Figure 3(c)***
>
> Thank you for raising this point. Figure 3(c) is intended to illustrate the high lexical diversity present in our dataset. The inner ring represents the first word in each opening sequence, while the outer ring represents the word that follows. The abundance of very thin segments indicates that no single word or phrase is disproportionately overrepresented. This high diversity prevents models from overfitting to specific lexical patterns and creates a more challenging evaluation setting.
>
> Furthermore, many of the opening sequences are questions (e.g., “How” and “Why”) rather than declarative statements, reflecting the inherent ambiguity and conversational nature of real-world online discussions. This aligns with our goal of modeling noisy and uncertain retrieval contexts.
>
> Both of these points (lexical diversity and prevalence of questions) are discussed in Section 2.3 (lines 181–183).
> Regarding the concern that the figure may appear “corrupted,” the large amount of white space is simply an artifact of the extreme diversity: the segments in the outer ring are so numerous and thin that they visually blend together. To prevent confusion, we will revise the caption in the final version to clarify these points.

---

> > ### Comment · Reviewer_sXya · 2025-08-05
> >
> > Thank you for your response. I have no further concerns.

---

> > > ### Author Response · Authors · 2025-08-07
> > >
> > > Thank you very much for reviewing our paper and recognizing our work. We are glad that our clarification addresses your concerns.
> > >
> > > We will include the discussed points in the final version.

---

### Note · Authors · 2025-08-12

We thank all four reviewers for their time, careful reading, and constructive engagement. We are encouraged that they consistently recognized the novelty and value of RAGuard as the first benchmark to evaluate RAG robustness against *misleading* retrievals in a real-world political fact-checking setting. Reviewers highlighted:
- **Real-world sources** (*mwAZ, xwdU*) over synthetic edits
- **Importance of the political domain** (*mwAZ*) for its noisy, evolving nature
- **Clarity of taxonomy** (*Ysze*) of document categories
- **Breadth of evaluation** (*sXya*) across leading LLMs
---
By the end of the discussion phase, our rebuttal clarifications had addressed the main concerns. Key points and new evidence:
- **Research Focus** (*mwAZ*) – RAGuard is the first benchmark to study *misleading* retrieval in political fact-checking. Unlike prior *noisy*-retrieval work (e.g., HotpotQA distractors, CRAG API noise), our dataset includes plausible yet deceptive Reddit content requiring nuanced reasoning to avoid being misled, addressing an underexplored RAG vulnerability.
- **Task Framing** (*sXya, mwAZ, xwdU*) – The task is *fact-checking* against PolitiFact gold verdicts, not misinformation detection. GPT-4–derived “supporting/misleading” labels are used only for intermediate analysis, never for training or final evaluation.
- **Annotation Reliability** (*sXya, mwAZ, xwdU*) – Re-annotation of document labels with Claude 3.5 Sonnet and Gemini 1.5 Flash found substantial (κ=0.789) and moderate (κ=0.650) agreement, confirming patterns are not unique to GPT-4.
- **Expanded Experiments** (*sXya, mwAZ, Ysze*) – Added to test vulnerability across model scales, evaluate robustness methods, and assess non-misleading noise:
  - **OLMo-1B**: Vulnerability extends to smaller models.
  - **Corrective RAG**: Robustness methods, while effective against some noise, remain vulnerable to real-world misleading retrieval.
  - **Unrelated Evidence**: Even topically related but uninformative documents can mislead, showing corpus noisiness beyond misleading cases.
---
**Final Paper Updates** — We will:

1.  highlight the research focus and taxonomy in the abstract;
2.  emphasize fact-checking framing in the Dataset section;
3. include new agreement studies; and
4. update results with added baselines.

We appreciate the reviewers’ engagement and believe these revisions both resolve raised concerns and significantly strengthen our contribution to RAG evaluation research.

---

### Decision · Program_Chairs · 2025-09-18

**Decision:**

Accept (poster)

**Comment:**

This paper addresses the gap in RAG evaluation by focusing on robustness against misleading retrievals, a realistic but understudied challenge. noting that it is the first benchmark to evaluate RAG robustness against misleading retrievals in a real-world political fact-checking setting. All ratings are positive, and most of the reviewers’ concerns were satisfactorily addressed in the rebuttal. However, the authors are encouraged to incorporate the reviewers' suggestions in the final version.